# Sources and preferences for nutrition information among older adults: A scoping review

Jane McClinchy[1]*, Angela Dickinson[1], Emily Barnes[1], Tai Ibitoye[1], John Jackson[2], Amander Wellings[2]

1 School of Health, Medicine and Life Sciences, University of Hertfordshire, Hertfordshire, United Kingdom, 2 Public Involvement in Research Group, University of Hertfordshire, Hertfordshire, United Kingdom

* j.1.mcclinchy@herts.ac.uk

## Abstract

A nutritionally adequate diet is essential for older adults to support healthy ageing and reduce the risk of malnutrition. With over a million older adults in the UK affected or at risk, understanding where they source nutrition information is critical for designing effective public health interventions. This scoping review mapped existing studies on the sources and preferences for nutrition information among older adults. A comprehensive search of PUBMED, Scopus, and CINAHL (March 2023; updated February 2025) yielded 8936 records, of which 15 studies reporting on 14 research projects met inclusion criteria. The majority of studies reported on multiple sources including magazines, family and friends, television, dietitians, general practitioners, internet and embodied knowledge (hidden and unconscious gained from personal experience). Educational level, gender, and trust were found to influence uptake and use. Further research is needed to assess the impact of these information sources and identify strategies to support older adults in making informed food choices that promote healthy ageing.

## Introduction

### Malnutrition in older adults in the UK

Healthy eating guidance for the general population in the UK is encapsulated in the Eatwell Guide [1]. However, this guidance is not considered by some to be sufficiently specific to be applied to older adults due to changes in nutritional requirements as a result of ageing [2]. The focus on the health of older adults is now more relevant due to population ageing across the world [3] as well as in the UK [4]. In England and Wales, the number of older adults (aged 65 and above) has increased from 9.2 million (16.4% of the population) in 2011, to 11 million (18.6% of the population) in 2021 [4]. Globally by 2050, 1 in 6 of the population will be aged 60 and above [3].

**Data availability statement:** All relevant data are within the paper and its Supporting Information files.

**Funding:** Initials: JM, AD, Grant number: Award Reference RCP1007006, Funder Reference BB/W018349/1 Full name of each funder: Biotechnology and Biological Sciences Research Council Food for added life years: Putting research into action (Food4Years) grant. URL: https://gtr.ukri.org/projects?ref=B-B%2FW018349%2F1 The funders did not play any role in the study design, data collection and analysis, decision to publish, or preparation of the manuscript.

**Competing interests:** The authors have declared that no competing interests exist.

Although energy requirements in older adults are often lower compared to the general adult population due to changes in body composition, requirements for vitamins and minerals remain the same apart from iron which has a lower reference nutrient intake [5 p29]. Additionally, recommendations for protein intake may increase for older adults to prevent sarcopenia [6]. The consumption of a healthy and nutrient dense diet is important in older adults to ensure healthy ageing [2,3,7]. Physiological and psychological factors commonly associated with ageing such as loss of taste and smell, depression and social isolation may result in loss of appetite [8] impacting on the ability of older adults to meet their nutritional requirements which can then result in the development of malnutrition [9].

Malnutrition is normally defined through a two-stage approach initially using a validated risk tool followed by a phenotypic assessment (weight loss, low BMI) and aetiologic assessment (loss of appetite, feeling weak or tired, increased incidence of episodes of illness) [10]. Before the Covid-19 pandemic the number of older adults living in the community in the UK with malnutrition or undernutrition was estimated to range from between 5 and 10% [11,12], affecting over 1.3 million adults in the UK alone. As well as impacting quality of life in this demographic, this has a significant socioeconomic burden.

Malnutrition was estimated to cost the UK £23.8 billion in 2017 [13,14]. Current societal challenges including the cost-of-living crisis mean that the number of adults living with malnutrition is likely to have increased [15]. Data from the Health Survey for England show that 79% of adults aged 65–74 years and 69% aged 75 years and over are overweight or living with obesity [16].

Concerted effort is needed to address malnutrition in older adults in order to address this public health crisis. Malnutrition is of global concern and part of the wider public health agenda where there is a focus on ending hunger and ensuring healthy lives for all ages [17]. While there has been a reduction in hunger and mortality rates since 2015, the gains in these areas have declined since the pandemic [18]. In the UK, 1 in 10 aged 60 and above (1.4 million) have been finding it more difficult to access healthy food since the pandemic [19] and almost half of those screened for malnutrition have been found to be malnourished [20]. There is a UK public health focus on addressing malnutrition [21] requiring actions to prevent, recognize and manage malnutrition once identified [8]. Research suggests that older adults have a low awareness of malnutrition and so part of the effort to address malnutrition with older adults include nutritional awareness [22]. There is a need for nutritional education and/or interventions focused on increasing the nutrient density of foods chosen by groups of the older population, including those who are malnourished and/or frail. In order to support nutrition strategies to ensure healthy ageing, there is a need to understand the information practice of older adults [23]. That is, where they source nutrition information, how they understand and respond to information, and how this information can be used to support adults to make the dietary changes necessary to meet changing nutritional needs.

### Nutrition education resources used by older adults

Health information practice is a key component enabling self-management to maintain health [23], however the process of information practice and information

gathering is instinctive and taken for granted [24]. Although, people are exposed to a range of sources of nutritional information, including media sources (TV, radio, newspapers etc), the internet, as well as labels on food packaging [25]. Nutrition information is also gained through individual daily practices [26,27] leading to hidden tacit knowledge that is embodied [28]. However, we know very little about where older adults source nutritional information, or how this informs what they choose to eat. While current front of pack food labels is offered as a source of nutrition information for older adults [29], the population-based approach of providing advice from statutory sources, often focusing on addressing obesity, can impact negatively on those who are nutritionally vulnerable [29].

Nutritional labelling in the UK should comply with regulations set by Government [30] however the current traffic light system is voluntary [31]. The aim of nutrition food labelling is to 'ensure that consumers understand what they are buying and that "it is what it says it is"' [32]. Although food labels have been found to be one of the main sources of nutrition information used by the public [33], the use of food labels by older adults is low [29,33]. Research suggests this group may have difficulties in interpreting the information [29,33,34] making food labels potentially ineffective in facilitating food choices and potentially leading to a worsening nutritional intake. There is anecdotal evidence that there may be unintended consequences of the "one-size fits-all" approach to nutritional labelling [26] by those working with older adults. The quote below identifies the concern about the inappropriate focus on foods labelled red (high) amongst older adults resulting in weight loss and subsequently malnutrition:

> *'Many of the people attending our services worry about eating things that may be 'bad' for them, such as those marked with a danger-invoking, red traffic lights. Sadly, many of these people join our service having lost weight unintentionally, and at risk of, or already malnourished.'* (Sarah Wren, Chief Executive of Health and Independent Living Support (HILS) - a social enterprise that supports older people with meals on wheels, nutritional advice, and wider health and wellbeing services.)

Maintaining a nutritious, balanced and enjoyable diet plays a crucial role in ageing healthily and avoiding conditions that arise from malnutrition [2]. As the world's population continues to age, understanding what barriers older adults face in accessing healthy meals requires more focus. A systematic review undertaken by Host et al. [35] identified that there are a range of factors influencing nutritional intake in older adults. While Brownie [36] found that information about what to eat has a key influence on food practices, the impact of being on the receiving end of nutritional campaigns over a longer period of time may make decisions about what to eat more challenging.

Recognizing what messages concerning a healthy diet are being received by older adults through their information practice [23] and where older adults find such information in the first place might give us an insight into how decisions about food are being made by this population group.

Although there is research exploring sources of health information in healthy older adults [37] and in those with long term conditions [38] as well as sources of nutrition information in the general public (for example [39]), there appears to be limited research exploring sources of nutrition information and their impact on food decisions in older adults. There is no one solution to reducing malnutrition among older adults, however, influencing adults' eating habits through provision of nutritional information is worthy of exploration. Obtaining accurate and suitable nutrition information is fundamental to informing healthier dietary choices, positive nutrition attitudes, and optimizing nutritional status.

This study formed the first phase of a two-stage project entitled: "Exploring sources of nutritional information used by older people: A feasibility study". The second phase involved the development of a nutrition information diary [details available in S1 File: Food4years conference]. The project team was supported by an advisory group including members from the community working with older adults and members of the University's Public Involvement Research Group (PIRg) [40]. Bimonthly meetings were held to monitor progress and to ensure that the project reflected community needs of older adults and lay views and opinions. Although the UK uses the age of 65 when referring to older adults [4], we have used the age range 60 and above to refer to older adults. Early in the research process we found that there was limited

research on the topic of nutrition information in older adults in the UK. Therefore, the focus of the search included studies undertaken outside the UK where many used the age of 60 and over to define older adults, a definition also used by the WHO [3].

### Aim

This study aims to explore sources of information free living older adults draw on to inform their food choices and understanding of a healthy diet using health information practice as the conceptual framework.

## Materials and Methods

A scoping literature review following the methodology of Arksey and O'Malley [41] incorporating additional details developed by Levac et al. [42] was undertaken in order to map the breadth of academic evidence currently available and identify gaps in the research evidence pertaining to nutritional information used by older adults to inform their eating habits and wider notions of a healthy diet. Scoping reviews are undertaken to assess and understand evidence gaps and differ from systematic reviews which focus on answering very specific research questions [43,44]. Peters et al. [45] argue that regardless of approach, all types of evidence synthesis should be undertaken in a systematic manner and follow methodological guidelines. Therefore, this scoping review was carried out following the six stages initially set out by Arksey and O'Malley [41,46]: identifying the research question, identifying relevant studies, study selection, charting data, and collating, summarizing and synthesizing results, consulting with experts and end-users (see Fig 1). We undertook a quality assessment for each study using a mixed methods assessment tool which can be used across a range of methodologies [47] [S2 File]. No studies were excluded following this process [45]. The checklist on reporting scoping reviews [43] was completed to facilitate rigor [S3 File].

### Criteria for inclusion and search strategy

Inclusion criteria are summarized in Fig 2. To be eligible for inclusion, studies had to include older adults aged over 60 years. Where studies included a range of participants across age ranges, data needed to be reported separately for participants over 60. The review focused on free living older adults and does not include studies which focus solely on older adults living in care homes, or those with particular diseases or long-term medical conditions as the findings would not be generalizable to the wider population [48]. Participants had to have been exposed to or impacted by at least one source of nutrition information. Eligible papers report outcomes referring to a change in food habits or preferences or a change in the understanding, perception or awareness of nutrition information. Papers were excluded if they focused on younger adults alone, did not relate to sources of nutrition information among older adults; were reviews, a letter to the editor, conference abstract; or if no full text was available. In order that the review reflected current issues, papers were excluded if they were published before 2000. Studies written in languages other than English were excluded.

The search was conducted in March 2023 (search 1) and updated in February 2025 (search 2), across three databases: PUBMED, Scopus and CINAHL. Predefined keywords in the search strategy based on the Population, Exposure, Outcomes (PEO) framework as outlined in Fig 2 were agreed with the research team and at the public advisory group meetings. The search strategy was developed with the assistance of an information manager specialist in the Library and Computing Service at the University of Hertfordshire. The results of search 1 were downloaded into Rayyan screening software [49], duplicates removed, and two assessors (TI, EB) independently evaluated the relevancy of each paper against the inclusion/exclusion criteria (that is each paper was evaluated by each assessor). The resulting studies were then downloaded into Microsoft Excel. Due to the small number of results from search 2, these were downloaded directly into Excel, duplicates were identified and removed by hand by two members of the research team (JM,AD). As the initial search team were not available to undertake this

| Steps | |
|---|---|
| 1. Research question identified | • A research team was created involving subject specialists, supported by an advisory group (members of the community working with older adults and lay members of the university's Public Research Involvement group).<br>• Using the Population, Exposure and Outcome (PEO) framework the research focus was developed with research team. |
| 2. Identification of relevant studies | • A librarian experienced in undertaking literature searches provided guidance on realising the research question into search terms.<br>• These were discussed with the research team<br>• Three databases were searched using the agreed search terms with English language and the year 2000 and later as filters |
| 3. Selection of studies to be included | • All study designs were considered for inclusion<br>• All studies that met the inclusion criteria were included<br>• Two reviewers screened full texts for inclusion. A third reviewer who was a member of the Public Research Involvement group and who was trained in scoping reviews settled disagreements.<br>• Regular team meetings were held to monitor progress of study selection and data charting |
| 4. The data was charted | • The research team created a data extraction tool (see data charting process).<br>• Charting was undertaken and agreed by two members of the research team and checked by the the lead author (JM)<br>• Quantitative and qualitative data were recorded |
| 5. The data was synthesized | • Quantitative data was summarized.<br>• Qualitative thematic analysis was undertaken to identify themes.<br>• Themes were iteratively developed as data progressed.<br>• Main themes were discussed with the research team.<br>• A quality assessment was undertaken on all studies<br>• Tables were prepared to present the charted data.<br>• Stages 2-4 were repeated |
| 6. Experts and end-users were consulted. | • Consultation with the research team and advisory group on the findings following data synthesis |

**Fig 1. The research process.**

task, two assessors (JM, AD) independently evaluated the relevancy of each paper against the inclusion/exclusion criteria. A third reviewer who was a member of the PIRg and trained in scoping reviews (JJ) was asked to settle any differences of opinion across both search 1 and search 2 to ensure consistency. Agreement was reached for all studies included in the final review.

| Search strategy | Inclusion criteria description | Exclusion criteria | Search terms |
|---|---|---|---|
| Publication type | Published research articles in English from 2000 | • Publication not in English<br>• Published prior to 2000<br>• Full text not available | |
| Study design | • Meta-analysis and systematic reviews<br>• Randomised control trials<br>• Cohort studies<br>• Case-control studies<br>• Cross-sectional studies<br>• Qualitative studies | • Not reporting on primary research<br>• Reviews<br>• Letter to the editor<br>• Conference abstract | |
| Population | Men and women aged 60 years and above, who live in the United Kingdom. This was expanded to international as few studies involved older adults who lived in the UK. | Studies focusing on younger adults alone. Studies that focus on those living in long term care facilities alone. | "UK older adults" OR "older adults" OR "older men" OR "older women" OR "older people" OR elderly OR "older person" OR senior OR "senior citizens" OR geriatric OR "over 60 years" |
| | | | AND |
| Exposure | Sources of nutrition information, such as but not limited to food or nutrition labels, Film/TV, internet, healthcare professionals, books, magazines, friends, family. | The focus of the study did not relate to Nutrition Information | "Sources of nutrition information" OR "nutrition information" OR "food label" OR "food labeling" OR "nutrition label" OR "nutrition labeling" OR OR (nutrition information) AND (books) OR (magazines) OR (newspapers) OR (TV) OR (internet) OR (social media) OR (professionals) OR (friends) OR (family) OR (relatives) OR (emotion) OR (feeling) OR (mood) |
| | | | AND |
| Outcome | At least one of the following:<br>• understanding of food labels,<br>• knowledge of food/nutrition<br>• food intake<br>• food habits<br>• food choices<br>• attitudes towards food<br>• perception towards food<br>• food preferences<br>• food shopping<br>• cooking<br>• nutrition awareness<br>• nutrition education<br>• beliefs | No outcome reported on the impact of nutrition information | "food intake" OR "dietary intake" OR "food habits" OR "health habits" OR "food choices" OR "food preferences" OR "nutrition attitudes" OR "nutrition perception" OR "nutrition awareness" OR "nutrition education" OR "nutrition knowledge" OR "food knowledge" OR "beliefs" OR "shopping" OR "cooking" |

**Fig 2. Inclusion and exclusion criteria and search terms.**

## Data charting process

Data from each study were extracted and charted into Excel by the two members of the research team who undertook the search (search 1: TI,EB, search 2: JM,AD) using headings: author, year, aims of the study, country and study setting (for example community or hospital), study design and methods, number, age and gender

of participants and finally the outcome of the research and what the study adds to what is known about nutrition information sources used by older adults. Where the study was undertaken in the hospital setting, the paper was reviewed to confirm that the participants were free living. Charted data was checked by the lead author (JM). The studies were grouped into methodologies of quantitative, qualitative and mixed methods. This final synthesis was discussed with the research team and advisory group.

## Results

The initial search on Rayyan yielded 8,864 results, 45 duplicates were identified automatically by Rayyan, leaving 8819 unique entries. The updated search undertaken using Excel revealed a further 120 results. Hand searching identified three duplicates. Overall a total of 8936 unique entries were identified. Titles and abstracts were screened against the inclusion criteria. No further duplicates were identified. From these results, 60 papers were identified for a full text review. An additional 45 papers were excluded as they failed to meet the inclusion criteria, leaving 15 papers to be included in the final review. Although two papers report on the same study [50,51], they have both been included as they report on different aspects. The number of studies identified at each stage is given in Fig 3, and the studies are summarized in Figs 4–7.

### Characteristics of included studies

**Study designs.** Included papers are comprised of qualitative (n = 5), quantitative (n = 8) and mixed method (n = 2) designs. All of the quantitative papers [53–60] are cross-sectional studies and most record the levels of knowledge or understanding that population groups have regarding nutrition information using survey methods. Only one study involves delivery of an intervention [55]. Three of the qualitative studies used focus groups and interviews [61–63], one [64] used just interviews and one used discussion groups [65] to explore outcomes associated with attitudes towards and perceptions of sources of nutrition information. The two mixed method papers [50,51] combine quantitative methods with further elaboration from participants through focus groups.

**Settings.** Studies were conducted in Canada (n = 4) [50,51,60,63] 2 of these papers report on the same study [50,51], United States of America (n = 3) [53,57,61], Australia [65] (n = 1), Chile [53] (n = 1), Singapore [59] (n = 1) and European countries of Sweden (n = 1) [62], Portugal (n = 1) [58], Italy (n = 1) [56], Germany (n = 1) [64], and one study [55] was undertaken across five different European countries (France, Italy, Poland, the Netherlands and United Kingdom (22% of the 1144 participants were from the UK)).

The majority of studies recruited participants from community settings (n = 7) [53–55,57,60,62,63], centres for older adults such as town halls or retirement villages (n = 4) [50,51,61,64], while three recruited participants from hospitals (n = 2) [56,59] and healthcare centres [58], and one recruited participants online [65].

**Participants.** *Age:* The age range of participants across the studies was 50–103 years. Although the inclusion criteria for studies was to involve adults aged over 60, two studies involving participants under the age of 60 were included as the mean age was over 60 [57,65]. McKay et al. [57] involved adults aged between 50–87 (mean age 64.6) and Turner et al. [65] involved participants aged 56–74 (mean age 63.3).

*Gender:* In one study [62] all participants were women. There were no studies where all participants were men. One study [57] included 50.6% women and 49.4% men. The overall distribution of participants between women and men was 63% women 37% men.

### Sources of nutritional information

The studies found that a wide range of sources of nutritional information were used by older adults. The most common sources were magazines which were identified by eight studies [51,55–57,59–61,64]; family and friends (seven studies) [51,54,57,59,61,62,64] television (six studies) [54,56,57,59,61,64]; dietitians (six studies) [51,54–56,59,61]. Most studies reported on several sources of information, 4 studies reporting on a wide range of sources [58,59,61,64] and 4

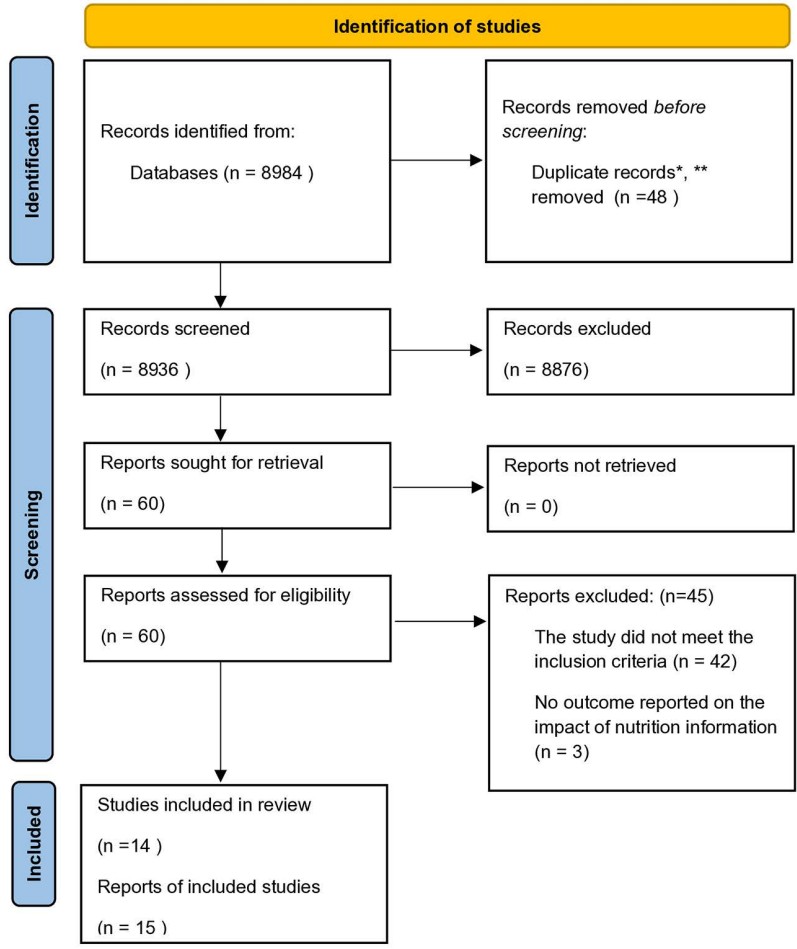

**Fig 3. Prisma 2020 flow diagram for the scoping review (adapted from Page et al. [52]).** This work is licensed under CC BY 4.0. To view a copy of this license, visit https://creativecommons.org/licees/by/4.0/.

studies reporting on one source. Doma et al. [50] and Mccharles and Fox [63] who reported on national food guidance, González-Contreras et al. [53] on food labels and Turner at al [65] who reported on the internet as a source of nutrition information. These are summarized in Fig 8

## Outcomes of studies

In this section we explore the main themes identified through the analysis of the studies. The studies in alphabetical order are collated by main theme in Fig 9: Sources relied on and preferred; Need for information; Embodied nutritional knowledge; Impact on food practices; Gender differences; Education levels and skills; Trust and conflicting messages. The subthemes by study are listed in Fig 10.

| Author Year | Study Aims | Country and Setting | Study design and Method | Participants | Outcome of the research and what the study adds to what is known about nutrition information sources used by older adults |
|---|---|---|---|---|---|
| González-Contreras et al. [53] 2024 | To determine the association between the frequency of warning labels reading (WLR) in foods, with dietary patterns and body mass index in older adults (OA) | Chile Community | Quantitative, secondary data analysis of data collected from the National Health Survey 2016 2017 | 1510 older adults over the age of 60 who took part in the survey. 64.4% women, 35.6% men | • 53.9% of the participants read warning food labels<br>• 67.11% of women and 32.89% of men read warning labels.<br>• Overall 86.39% of those who read warning labels lived in urban areas while 13.61% lived in rural areas (p<0.001).<br>• A higher percentage of those with a higher education although a small sample were more likely to read food labels than those with a low education (56.4% versus 33%).<br>• Those who read warning labels were more likely to meet nutritional recommendations for fish (p<.01), dairy (p<0.001), whole grain cereals (p<0.001), pulses (p<0.05), consumption of fruits (p<0.01), consumption of sugary drinks (p<0.001) and water (p<0.001), but not vegetables (p=0.167, sugar sweetened juices (p=0.382). |
| Heuberger and Ivanitskaya [54] 2011 | To examine the preferred sources of nutritional information by younger and older people. | USA Community | An online information literacy assessment of younger adults (study 1) and interview surveys with older adults (study 2). | Study 2 included 1,100 older adults aged 60-103. Majority female (65%). 55% were aged 60-75, 40% were aged 75-90 and 5% older than 90.<br><br>(Study 1 included 1680 adults under the age of 25, 71% female ) | • Preferred sources of information varied by age group.<br>• Younger participants (study 1) preferred Dietitians and the internet, while older adults who were in good health (Study 2) preferred health professionals while older adults in poor health preferred Dietitians.<br>• Older adults with lower education levels also preferred health professionals as a source.<br>• Family and friends were preferred by younger adults and older adults who were in good health.<br>• The internet was preferred by younger adults as opposed to older adults.<br>• There were no gender differences for older adults relating to source while younger male participants preferred the internet. |
| Jeruszka-Bielak et al. [55] 2018 | To investigate whether nutrition related knowledge and attitudes were associated with lifestyle and health features among older adults from five European countries (France, Italy, Poland, the Netherlands and United Kingdom). | Europe France, Italy, Poland, the Netherlands and United Kingdom Community | Healthy older volunteers were randomly assigned to two groups: intervention (NU-AGE diet) or control.<br><br>NU-AGE diet group received monthly education sessions on the Mediterranean diet. These were also supported by the provision of foods encouraged on the diet. With 12 month follow up. | 1,144 healthy elderly volunteers (65–79 years) N= 635 (women) | • Good nutritional knowledge was associated with reduced BMI and higher physical activity, younger age, higher finances and education and was higher in the intervention group.<br>• No difference between genders.<br>• Positive nutritional attitudes were associated with lower BMI, self-reported good appetite and being Italian.<br>• Higher self- evaluated nutritional knowledge was associated with more positive nutrition attitudes. |
| Laurenti, et al. [56] 2020 | To investigate older adults' demographic and socio-economic characteristics, knowledge, attitudes and practices in relation to food safety and healthy diet. | Italy Community dwelling but recruited whilst attending a clinic or hospital | Questionnaire | 201 adults over 65 years old. n = 51 (men) n = 150 (women) | • While almost all respondents reported they read food labels including the expiration date, a number did not follow safe food hygiene practices.<br>• 49.1% defrosted food at room temperature, 33.8% consumed out-of-date food and 37.8% kept leftovers in the fridge for more than 5 days. |

**Fig 4. Summary of included studies: Quantitative studies 53-56.**

## Sources relied on and preferred

A number of studies highlighted a wide range of preferred sources for nutritional information for older adults. Duerr [61] who used focus groups to explore information sources currently used and what learning methods were 'liked' by 37 participants grouped preferred sources into written mixed mediums, people, organisations, other (fairs, cooking demos, courses). Oliveira et al. [58] found preferences for a wide range of media, such as booklets with pictures, informational posters, and food education and practical cooking sessions. While participants in Ong et al. [59] reported that television (40%) and internet (40%) were the most used sources followed by printed materials such as newspapers (39.3%) books/ magazines (32.3%) and word of mouth, e.g., through friends (20.3%) and family (14.3%). McKay et al. [57] also found newspapers (61.8%) as being frequently referred to alongside doctors (61,8%), followed by magazines (60.1%) and television (49.1%). However, Heuberger and Ivanitskaya [54] found that most participants preferred to receive nutritional information from health care professionals (HCPs) who were not registered dietitians and Maccharles and Fox [63] found that their participants did not feel that guidance from HCPs would be helpful.

Studies explored preferred sources of information about specific food items. Farrell et al. [51] found the most common preferred sources for bean information included food labels (54.8%) as well as brochures (51.2%) and the internet (47.2%). While Vella et al. [60] explored sources of information about functional foods with 200 older adults (70% women) through a researcher completed questionnaire and found that 68.5% preferred newspapers, magazines, and books as sources of nutritional information about functional foods.

| Author Year | Study Aims | Country and Setting | Study design and Method | Participants | Outcome of the research and what the study adds to what is known about nutrition information sources used by older adults |
|---|---|---|---|---|---|
| McKay, et al. [57] 2006 | To examine specific sources of nutrition information among an older adult population, and compare the difference in sources associated with extent of education. | Boston (USA)<br><br>Community | Mail survey designed to test the efficacy of nutrition messages.<br><br>Participants were asked to select from a list the sources they primarily rely upon for information about nutrition | 176 adults. The age range of study participants was **50 to 87 years**<br><br>Mean age 64.6yrs<br><br>49.4% men<br>50.6% women. | • Those who were less educated relied significantly more on Doctors, TV and neighbours for information and more likely to read magazines.<br>• Older adults accessed information in written form, and women were more likely than men to use friends as a source of information. |
| Oliveira, et al. [58] 2021 | To study the perceived need and preferences regarding sources of information about healthy eating among older adults and socio-demographic characteristics. | Portugal Primary Care health center | A structured questionnaire developed within a larger project which aimed to reduce undernutrition in older people | 602 older adults (≥65 years old)<br><br>N=325 (women)<br>N = 277 (men) | • 69.3% would like to receive more healthy eating information.<br>• Most preferred audiovisual or leaflets with images or text.<br>• Women preferred practical information such as cooking sessions or films.<br>• Those with higher education levels or greater independence preferred practical as audiovisual or leaflets with images or text. |
| Ong et al. [59] 2021 | To understand the nutrition knowledge, competencies and attitudes of community-dwelling older adults in Singapore. Part of the Strengthening Health In ELDerly through nutrition (SHIELD) study. | Singapore<br><br>Community dwelling recruited through attendance at hospitals and clinics | Questionnaire using a locally developed and validated scale was used to measure nutrition knowledge, competencies, attitudes and sources of nutrition information. | 400 participants<br>N= 183 (men)<br>N= 217 (women) | • While females in the study had higher nutrition knowledge index scores, males tended to leave decision making up to others, most reported nutritional values influenced their food choices.<br>• Common sources cited in this order Television (40%), internet (40%), newspapers (39.3%), books/magazines (32.3%) friends (20%) and family (14.3%).<br>• Compared to males, females were significantly more likely to turn to TV (45.2% vs. 33.9%), radio (19.8% vs. 8.7%) and friends (24.4% vs. 15.3%) for nutrition information. |
| Vella, et al. [60] 2014 | To identify the need for information related to functional foods among older adults and to assess awareness and perceptions of health claims on functional food | Canada<br><br>Community | Researcher administered questionnaire. | 200 older adults >60 years<br><br>Mean age of 70.8 | • 93.0%. consumed functional foods.<br>• Increased awareness and knowledge were the main factors considered to promote consumption (85.5%). 63.5% wanted more information about functional foods, preferred sources newspapers/magazines/books (68.5%) and food labels (66.1%).<br>• 93.5% aware of health claims. Those with more education were more likely to report being aware of health claims (p = 0.045). |

**Fig 5. Summary of included studies. Quantitative studies 57-60.**

Sources for nutrition information were selected for particular reasons. Duerr [61] found that participants chose a source as it was an effective way to learn and both Duerr [61] and Rueter et al. [64] found that enjoyment and interest on a topic influenced using a source of information. The availability of information also influenced use. Duerr [61] found that being able to take information away and read it later and both Rueter et al. [64] and Maccharles and Fox [63] found that having information available at the point of purchase in the form of a food label would be effective in guiding them whether to purchase a food or not.

The situation of participants was also found to influence preference for sources of information. Heuberger and Ivanitskaya [54] found that the presence of poor health meant they preferred advice from a dietitian whilst those in good health preferred nutritional advice from a HCP who was not specialised. Oliveira et al. [58] found those with adequate social support preferred booklets with text.

Preferences for format of the information was identified by some studies. Duerr [61] found their participants valued practical learning methods such as demonstrations, discussions, classes, eating and tasting and Oliveira et al. [58] for practical cookery sessions as well as 'audiovisual' or leaflets with images alongside the text. Turner et al. [65] in their participatory design workshops explored preferences for the delivery of digital nutrition information. They found that while participants made use of a range of electronic devices including SMART phones, they preferred a web site as opposed to an app for nutritional information.

## Need for information

Eight studies reported on their participants' need for nutrition information. Oliveira et al. [58] who undertook a questionnaire with 602 older adults in Portugal found that most participants were concerned about healthy eating (87.5%) and would like more information on this topic (69.3%). However, two studies found that participants would like a broad range of nutrition information. Duerr [61] in their focus group study with participants in the USA identified seven categories that their participants wanted more information about: basic nutrition, diet and disease, lifestyle,

| Author Year | Study Aims | Country Setting | Study design and Method | Participants | Outcome of the research and what the study adds to what is known about nutrition information sources used by older adults |
|---|---|---|---|---|---|
| Duerr [61] 2003 | To assess the perceived nutrition education wants and needs of older adults. | Ohio (USA)<br><br>Senior centres, town hall and independent living facility. | Five focus groups, semi structured interview style. | 37 non-institutionalised adults aged 60-89<br><br>N= 28 (women).<br>N= 9 (men) | (1)Preferred information sources were grouped into five main categories:<br>• Written materials (e.g., newspapers, food labels, magazines, books),<br>• Mixed mediums (e.g., television, computers, grocery store samples/pamphlets),<br>• People (e.g., doctors, dietitians, friends, family),<br>• Organisations (e.g., the county Office for Older Adults, health department, churches).<br>2) Learning methods were identified through when participants said they "really like"…<br>These were grouped into 4 categories:<br>• Methods (e.g., demonstrations, discussions, classes),<br>• Written/physical sources (e.g., reading, handouts, tapes, cookbooks),<br>• People (e.g., experts in the field),<br>• Others a (e.g., learn by eating and tasting).<br>• Others b (e.g., fairs, cooking demonstrations, nutrition courses taken at the university<br>• Being able to keep information and review it in a written from after the event or with specific instructions were valued. |
| Gustafsson and Sidenvall [62] 2002 | To explore food-related health perceptions and food habits among older women. | Sweden<br><br>Community | Qualitative interviews and 3-day food diary | 18 women aged 65–88, living alone or cohabiting, who independently managed shopping and cooking. | • Women who were influenced by mass media felt frustrated about different messages and did not know who to believe.<br>• The main messages adopted were a fear of fat and of gaining weight or a desire to lose weight.<br>• Those with less knowledge found it difficult to understand the messages but felt it was important to be informed.<br>• Some had tried using olive oil as it had been recommended, they agreed it was healthy but did not like the taste. |
| Maccharles and Fox [63] 2019 | To explore the food guidance needs and wants of a group of seniors living in Antigonish, Nova Scotia | Canada (Nova Scotia)<br><br>Community | Focus groups;<br>Older adults were asked about their views on Canada's Food Guide (CFG) and the Brazilian Dietary Guidelines (BDG). | 12 participants over the age of 65 years, living independently | • Canada's Food Guide was identified as a trusted source of information and related well to the food groups and directive statements.<br>• Portion sizes were confusing and participants found it difficult to determine the recommended amount to eat. Advice on food choices was not seen as being realistic in terms of cost and availability.<br>• Saw little value in distributing healthy eating information through health professionals.<br>• The holistic nature of the BDG was appealing but guidance on processed food and social eating was not seen as relevant. |
| Rueter, et al. [64] 2020 | Explore how older people in rural communities in Bavaria, Germany, perceive their food environment. | Germany<br><br>Community | Semi-structured face-to-face interviews with senior citizens in five German communities. | 35 participants 65 years and over.<br>Average age=71.2<br><br>N= 29 (women)<br>N = 6 (men) | • Doctors, pharmacists nutritionists were sources of nutritional information that was trusted<br>• The lack of transparency about the production on food labels of some specialist foods restricted their use.<br>• Media coverage was reported to impact food choice, but the experience of contradictory information from different sources (tv magazines) caused confusion.<br>• Overall, a feeling of being deceived by food labels made it difficult to make informed choices. |
| Turner et al. [65] 2024 | To explore technology and use among older adults and qualitatively determine the content needs and design preferences for an online nutrition education resource tailored to older adult consumers in Australia | Australia | Online participatory discussion workshops | 20 participants **aged between 56 and 74** Mean age = 63.3. N=16 (women)<br>N= 2 (men) | 5 main themes:<br>• **Technology use among older adults**/Perceived competence, Preferred devices websites and apps, Barriers to technology use, Facilitators of technology use<br>• **Flexible access** - using at home, but smartphone always available<br>• **Information needs** -nutrition basics, practical information recipes<br>• **Personalised experience**- "One size does not fit all"<br>• **Content from a trusted source**- important to have evidence-based information |

**Fig 6. Summary of included studies: Qualitative studies 61-65.**

specific foods, supplements, general education. Turner et al. [65] who undertook participatory discussion groups online found that older adults would like nutritional information such as nutrient values, practical information including how to read food labels, finding seasonal healthy food, using leftovers and a source that could be personalized to their own needs as "one size does not fit all" (page 6). Duerr [61] also found the need for adapting recipes and on how to eat healthily on a fixed income. Maccharles and Fox [63] who explored the Canada Food Guide with a small group of older adults in focus groups (women n = 10, men n = 2) found their participants needed to know how to select affordable healthy food and needed more guidance on portion sizes as this was not clear. However, nutritional information was considered valuable in making food choices. Farrell et al. [51] and Vella et al. [60] who explored information needs about beans and functional foods respectively found their participants would like more nutrition information about these foods. While Rueter et al. [64] found that their participants would like information as they were interested in the topic, they did not always need nutrition information. Duerr [61] also found some of their participants did not need information, in this case it was because they had no food issues. Maccharles and Fox [63] found that not all information was considered relevant to their participants. For example, they had no need for information on processed foods as they 'did not eat that kind of food'.

Although older adults make use of a wide range of sources to support their decision making, they do not always have food issues requiring information and information about foods that they do not normally eat are not relevant to them.

| Author Year | Study Aims | Country and Setting | Study design and Method | Participants | Outcome of the research and what the study adds to what is known about nutrition information sources used by older adults |
|---|---|---|---|---|---|
| Doma et al. [50] 2019 | To explore older adults' awareness and knowledge of beans in relation to their nutrient profile and their potential to promote health through reducing chronic disease risk.<br><br>To examine awareness of beans in relation to governmental dietary guidance (Canada's Food Guide 2007) | Canada<br><br>Community | Qualitative mixed-methods approach<br><br>Phase one: study questionnaire on awareness of beans)<br><br>Phase two: focus groups to explore participants' awareness of nutritional value of beans. | 250 community-dwelling older adults who were ≥65 years old.<br><br>76% female. | • The majority of older adult participants were aware that beans are included in Canada's Food Guide with no significant difference between bean consumers and bean non-consumers (87.5% vs. 81.2%; p = 0.20).<br>• The prevalence of bean consumption was only 51.2%, despite the fact that almost all participants agreed that beans are a healthy food, and that bean consumption could improve their health (98%).<br>• The specific health areas that the majority of older adults thought bean consumption could improve included body weight management, constipation, diabetes and heart health. |
| Farrell et al. [51] 2019<br><br>This paper is the same study as Doma 2019. | To explore bean consumption among older adults in relation to health claims and other information sources | Canada<br><br>Retirement villages, senior centres, University. | A mixed-methods explanatory sequential study design<br><br>Researcher-administered questionnaire followed up with 10 semi-structured focus groups. | 250 older adults aged (≥65 years old<br><br>N= 190 (women)<br>N= 60 (men) | • The majority were aware of and read about the nutrient content, nutrient function, therapeutic and disease risk reduction claims of sources (table).<br><br><table><tr><td>Claim</td><td>Nutrient content</td><td>Nutrient function</td><td>Therapeutic value</td><td>disease risk reduction</td></tr><tr><td>Aware</td><td>94.4</td><td>64</td><td>79.6</td><td>77.2</td></tr><tr><td>Read</td><td>91.5</td><td>85.6</td><td>87.9</td><td>88.6</td></tr></table><br>• However, only 51.5% consumed beans and 46.8% said that health claims would increase their consumption.<br>• 50% had read nutrition information about beans most frequent were food labels (68.0%), magazines and/or books (59.2%) and brochures (48.0%).<br>• 72.0% would like more information, the most common preferred sources including food labels (54.8%), brochures (51.2%) and the internet (47.2%). |

**Fig 7. Summary of included studies. Mixed Methods studies 50,51.**

## Embodied nutritional knowledge

The studies appear to indicate that the relationship between embodied nutritional knowledge (hidden and gained through individual practices [27,28]) and food chosen/eaten by older adults is not straightforward. Gustafsson and Sidenvall [62] who undertook qualitative interviews and a 3 day food diary exploring food related health perceptions and habits with 18 women in Sweden found that their participants held embodied perceptions about food and health. Their participants believed that feeling healthy meant that you must be eating healthily, and that this was enhanced by consuming home cooked meals. However, not all their views were positive; they had an embodied fear of fat which led to cooking low fat meals.

Studies found an association between nutritional knowledge and eating practices and behaviours. Vella et al. [60] found 85.5% of their participants felt that increased awareness and knowledge about functional foods would promote their consumption. Jeruska-Bielak et al. [55] found that good nutritional knowledge was associated with a healthier body mass index and physical activity and positive nutritional related attitudes (which they define as emotions, motivations, perceptions and cognitive beliefs). Farrell et al. [51] also found a link between beliefs and food behaviours, finding that participants who had a greater belief in the impact of diet on health were more likely to take notice and list use of health claims as influencing their food choices.

Conversely, studies also found that general awareness of a food or food practice does not always promote healthy food practices. Doma et al. [50] who undertook a researcher administered questionnaire with 250 older adults followed by focus groups with 10 participants found that while 98% of participants were aware that beans were a healthy food item, only 51.2% were regular consumers. Laurenti et al. [56] who undertook a self-administered questionnaire with 201 older adults (women n = 150, men n = 51) who were attending a ward-based, outpatient clinic gym session for older adults in Italy found that while most participants said they followed safe food practices, 49.4% defrosted food at room temperature and 33.8% consumed food out of date more than once per month and 37.8% kept food as leftovers for more than 5 days.

## Impact on food practices

Studies explored both whether a specific source of information had an impact on food choice as well as participant's perceptions on whether a source would impact their decision making. Ong et al. [59] in their questionnaire study exploring nutritional knowledge, competencies and attitudes found that of their 400 Singaporean participants aged 65 and above,

| Source/study | 50 | 51 | 53 | 54 | 55 | 56 | 57 | 58 | 59 | 60 | 61 | 62 | 63 | 64 | 65 | T* |
|---|---|---|---|---|---|---|---|---|---|---|---|---|---|---|---|---|
| Magazines | | X | | | X | X | X | | X | X | X | | | X | | 8 |
| Family /Friends | | X | | X | | | X | | X | | X | X | | X | | 7 |
| Embodied knowledge | | X | | | X | X | | | X | X | X | | | | | 6 |
| Dietitians | | X | | X | X | X | | | X | | X | | | | | 6 |
| GPs/Doctors | | X | | | X | X | X | | X | | X | | | | | 6 |
| Television | | | | | X | X | X | | X | X | X | | | | | 6 |
| Internet | | | | X | X | X | | | X | | X | | | | X | 6 |
| Food labels | | X | X | | X | | | | X | X | | | | X | | 6 |
| Books/Recipes | | X | | | X | | | | X | X | X | | | | | 5 |
| HCPs** | | | | X | X | X | | | | | | | | X | | 4 |
| Newspapers | | | | X | | | | X | X | | X | | | | | 4 |
| Leaflets | | | | X | | | | X | | | X | | | | | 3 |
| Radio | | | | X | | | X | | X | | | | | | | 3 |
| Audio visual | | | | | | | | X | | | X | | | | | 2 |
| Health authority | | | | | | | | | | | X | X | | | | 2 |
| National food guide | X | | | | | | | | | | | | X | | | 2 |
| Practical | | | | | | | | X | | | X | | | | | 2 |
| Church | | | | | | | | | | | X | | | | | 1 |
| Mass media | | | | | | | | | | | | X | | | | 1 |
| Scientists | | | | | | | | | | | | X | | | | 1 |

* T-total number of studies referring to this source

** HCP – Health Care Professional

**Fig 8. Collation of sources of information identified by the studies.**

78.8% said that nutritional values of foods would influence their food choices. Jeruszka-Bielak et al. [55] found that participants who received counselling from a trained dietitian/nutritionist had an improved nutritional knowledge and attitude when compared to the control group in their study. While 71% of participants in Vella et al. [60] said that advice from a healthcare professional would help increase consumption of functional foods. Turner et al. [65] who was looking at online delivery of nutritional information found that if information is individually tailored for example by adults being able to enter

| | Sources relied on and preferred | Need for information | Embodied nutritional knowledge | Impact on food practices | Gender differences | Education levels and skills | Trust and conflicting messages |
|---|---|---|---|---|---|---|---|
| Duerr [61] | x | x | | | | | x |
| Doma et al. [50] | | | x | | | | |
| Farrell et al. [51] | x | x | x | | | | x |
| González-Contreras et al. [53] | | | | x | x | x | |
| Gustafsson and Sidenvall [62] | | | x | | | | x |
| Heuberger and Ivanitskaya [54] | x | | | | x | x | |
| Jeruszka-Bielak et al. [55] | x | | x | x | x | | x |
| Laurenti et al. [56] | | | x | | | | |
| Maccharles and Fox [63] | x | x | | | | | x |
| McKay et al. [57] | x | | | | x | x | |
| Oliveira et al. [58] | x | x | | | x | x | |
| Ong et al. [59] | x | x | | x | x | x | |
| Rueter et al. [64] | x | | | | | | x |
| Turner et al. [65] | x | x | | x | | x | x |
| Vella et al. [60] | x | x | x | x | | x | x |

**Fig 9. Studies in alphabetical order by main theme.**

their own data and receive feedback, this would be useful and avoid a "one size fits all approach". González-Contreras et al. [53] undertook secondary analysis of the Chilean National Health Survey to determine the association between reading food labels and whether respondents met nutritional guidelines. Although they found that those who read these labels met the nutritional guidelines for fish, dairy, whole grain cereals, pulses fruits, sugared drinks and water this was not the case for the consumption of vegetables and sweetened fruit drinks.

## Gender differences

Although Heuberger and Ivanitskaya [54] who undertook a survey with 1100 older adults (65% women) in the USA did not find differences in preferred sources of nutrition information between men and women, a number of other studies did find differences. Oliveira et al. [58] whose participants comprised 54% women and 46% men found that a higher proportion of women preferred to receive information through practical cooking sessions (38.5%) than men (28.6%) (p = 0.044). Ong et al. [59] found that women were more likely to use television, radio and friends as a source of nutrition information, and McKay et al. [57] who undertook their survey in Boston with 176 older adults (50.6% women and 49.4% men) also found that women relied on friends more often than men. However, this was in contrast to Jeruszka-Bielak et al. [55] who found that 39% of men preferred friends and family as sources of nutrition information versus 33% of women. This study also found that more women than men preferred books and magazines (61% vs 49%), while more men preferred internet (43% vs 35%). González-Contreras et al. [53] looked at gender differences in the reading of warning food labels in the Chilean National Health Survey and found although not significant (p = 0.095) that more women than men read food labels. Of the 1510 participants (64.4% women, 35.6% men) 54% read warning food labels. Of the men who took part 32.8% read food labels and of the women who took part 67.1% read the warning food labels.

| Sources relied on and preferred | | Impact on food practices | |
|---|---|---|---|
| Wide range of sources | [61],[57],[58],[59] | Knowledge of nutritional values | [59] |
| HCPs* as a source of nutrition information | [54],[63] | Nutritional advice from a registered HCP* increases knowledge | [55] |
| Preferred sources of information about specific food items | [51],[60] | Nutritional advice from a registered HCP* increases consumption of healthy foods | [60] |
| Effective way to learn | [61] | Tailored advice | [65] |
| Enjoyment or interest | [61],[64] | Reading food labels was linked to meeting nutritional guidance | [53] |
| Being able to read through later | [61] | **Gender differences** | |
| Available at the point of purchase –ie food labels | [63],[64] | No gender differences | [54] |
| Health status | [54] | Gender preference for practical information | [58] |
| Presence of a social network | [58] | Gender preference for mass media | [59] |
| Format-practical | [61],[58] | Gender preference for friends | [55],[57],[59] |
| Format-website vs app | [65] | Gender preferences for books and magazines | [55] |
| **Need for information** | | Gender preference for internet | [55] |
| Healthy eating to improve health | [58] | Gender preferences for using food labels | [53] |
| Broad range | [61],[65] | **Education levels and skills** | |
| Recipes | [61],[65] | More education prefer leaflets with text as well as audiovisual material | [58] |
| Affordability | [61],[63],[65] | Less education more likely to use professional sources | [54],[57] |
| Portion sizes | [63] | Reading food labels | [53] |
| Skills- reading food labels | [65] | Awareness and use of food claims | [60] |
| Nutritional information | [61],[51],[59],[65],[60] | Seeking and critically appraising information | [59] |
| Did not need information | [61],[63],[64] | Internet use | [59] |
| They were interested in the topic | [64] | Digital skills need to access online | [65] |
| **Embodied nutritional knowledge** | | **Trust and conflicting messages** | |
| Perceptions of health and healthy eating | [62] | Trusted dietitians, nutritionists | [61],[55] |
| Promotes healthy eating | [55],[60] | Trusted HCPs* generally | [55],[64],[60] |
| Nutrition related beliefs and practices | [50],[51],[55],[56] | Trusted regulatory framework /national guidelines | [51],[63] |
| | | Including links to scientific research | [65],[51] |
| | | Being up to date | [65] |
| | | Conflicting advice in mass media and health authorities led to mistrust | [62] |
| | | Lack of trust in TV, magazines as they were contradictory | [64] |
| | | Feeling being deceived by food labels | [64] |

**Fig 10. Subthemes by study.**

## Education levels and skills

Educational level impacted preferred sources and use of information. Oliveira et al. [58] found that those with a higher educational level preferred leaflets with text and audiovisual materials (p values 0.013, 0.027 respectively). While McKay et al. [57] found that those with more than 4 years of college education (n=92) were more likely to use newspapers with a higher reading age (the New York Times, Time, Newsweek) and national public radio. However, those with less than 4 years of college education (n=84) were more likely to use magazines (specifically Good Housekeeping), neighbours and doctors, although the figures were small. Heuberger and Ivanitskaya [54] found that a higher percentage of those with a lower level of education were more likely to refer to a professional source of nutrition information than those with a higher level of education (<12 years 73% of women, 74% of men, versus >16 years 67% of both women and men). González-Contreras et al. [53] found that those with a higher edcation level were more likely to read warnings on food labels and Vella et al. [60] found that those with a higher education level were more likely to be aware of food claims on food labels. Ong et al. [59] found that a higher educational level facilitated being able to seek and critically appraise information and being able to use the internet. Turner et al. [65] also found that skills and competencies impacted preferred sources of nutrition information. They found that there was a need for digital skills in order to be able to access information online. In the participatory study, discussions indicated that while participants were themselves confident with the use of

online sources, having the option to print off a hard copy of information may make the source more accessible to those who were not so confident. The hardware used to access the internet was also found to be impacted by skills and competencies. While mobile phones were found to be useful for their ready availability, lack of confidence in their use and the small screen size limited their usefulness for all older adults.

### Trust and conflicting messages

Trust in nutrition information and concerns about conflicting messages in sources was identified across eight of the 15 studies. Duerr [61] found that when consulting about nutrition their participants said they would go to the person they respected or who was recognized in their field. This was not necessarily doctors who they felt did not always know very much about nutrition. Jeruska-Bielak et al. [55] found that dietitians held a high level of trust (50%) however, this was alongside books and magazines on health (49%) and doctors (42%). Rueter et al. [64] found that their participants trusted health care professionals (HCPs) in general, and Vella et al. [60] suggest that trust in advice from HCPs impacts food choice (71% said advice on functional foods would help increase consumption). Nationally available nutrition information was found to be a trusted source by Maccharles and Fox [63] and Farrell et al. [51]. Although Farrell et al. [51] found that participants trusted the regulatory framework for nutrition information, they felt that the scientific information could be more clearly linked to the guidance to help with trustworthiness. Turner et al. [65] also found that having access to scientific information backing up the advice as well as ensuring the information was up to date would ensure that a source was trusted.

Lack of trust and a feeling of being deceived was identified when accessing sources of nutrition information. Gustafsson and Sidenvall [62] found that women who said they were influenced by mass media and health authorities felt frustrated by different messages about food and health and did not know whom to believe. Rueter et al. [64] who sought to explore factors that influenced food choice with 35 older adults through semi-structured interviews found that although endorsement of specific foods in mass-media, specifically television and magazines, has the potential to influence food choice, participants experienced contradictory information and a feeling of being deceived by information on food labels.

## Discussion

We believe this is the first scoping review bringing together academic literature focusing on nutrition information and older adults. The review found that older adults used a wide range of information sources, including food labels, written media, the internet, health care professionals, and family and friends and identified the positive ongoing impact of a nutrition education intervention on food practices. Use of nutritional information appears to be mainly affected by educational level but there are also gender differences with women appearing to be more engaged with nutrition information. The source of nutrition information was important, affecting levels of trust and needing to be relevant and accessible. The review also found a potential positive impact of practical delivery of nutrition education information through sessions involving recipe adaptation, cooking, and tasting the food.

Most of the studies were undertaken outside the UK, relying on questionnaires to collect data. One study included older adults in the UK. This was also the only study to report on an intervention [55].

The wide range of sources of nutrition information investigated and preferred by older adults identified in this current study is in common with other studies exploring both health and nutrition information practices [38,39,67]. While this suggests that nutrition information is widely available and accessible, this also indicates a challenge in ensuring sources are accurate, do not conflict and meet the needs of those using the information [68]. Some studies investigated preference for sources of nutrition information overall, others concentrated on one source of nutrition information [65] or nutrition information about one food item [51,60]. The focused research enabled investigation into how to encourage consumption of foods recommended in healthy eating guidelines [51,60] and how to ensure nutrition information formats meet the needs

of older adults. Nevertheless, the study of nutrition information remains a complex topic impacted by the ever-increasing sources of information and variety and availability of food [68].

Studies exploring the reasons for accessing nutrition information among adults have also found a range of reasons for accessing nutrition information. Vrinten et al. [69] found five broad motivations for accessing nutrition information: health management, affective needs (including enjoyment), cognitive needs (interest), social integrative needs, and personal identity. Similarly, in this current study enjoyment and interest were found to impact on the choice of a source of information [61,64]. Practical delivery of nutrition information was identified by Duerr [61], Oliveira et al. [58], Turner et al [65]. Although Vrinten et al. [69] excluded recipes as a source of nutrition information because they were not considered a direct source of nutrition information, McClinchy et al. [27] found the sharing of recipes and food provisioning practices to be key sources of nutrition information for their participants living with type 2 diabetes (T2DM). Adam et al. [70] in their delivery of an online cooking course demonstrated the popularity of practical delivery of nutrition information in order to manage health. Studies identified that older adults would like information on how to find affordable healthy food [61,63,65]. Although the implication from this may be that healthy diets are more expensive than less healthy diets, research suggests otherwise and concurs with this current study that guidance is needed on how to obtain and prepare affordable healthy meals [71].

The finding by González-Contreras et al. [53] that food label use amongst older adults is low has also been found by other researchers [33]. While Farrell et al. [51] found that older adults found them useful for highlighting health claims about the importance of beans and González-Contreras et al. [53] found that those who read food labels were more likely to meet the majority of the Chilean food guidelines, Rueter et al. [64] noted that their participants were concerned about the potential for food labels to deceive. Although a small study involving 100 older adults from one region in the USA, Jackey et al. [72] found that while usage was high, many were not able to correctly interpret the information available. Castelo Branco [73] in their Brazilian study involving 17 telephone interviews identified vulnerability in older adults being able to make use of food labels, however Mahoney et al. [74] in their study with 47 older adults (aged over 55, in Ireland) found that front of pack labelling was actively used in order to make decisions about what food to buy.

Magazines and television were frequently preferred sources identified in the studies. There is limited research identifying magazines as a specific source of nutrition information, however Wills et al. [75] found that magazines may not always contain robust evidence-based information about nutrition. The value of television as a potential source of information was found by Rivero-Jiménez et al. [76], where participants in isolated rural communities in Spain relied upon television as a source of social interaction.

With the expectation for the growing reliance on the need for digital skills in the UK to facilitate the use of the internet [77] and the increasing use of technological hardware by older adults [78], the internet as a source of nutrition information is worthy of further discussion. The internet has been found to be a commonly preferred source of nutrition information across countries and age ranges and those living with long term conditions [38,39]. Ruani et al. [39] in an international study involving 3487 members of an organization aimed at those with an interest in learning about nutrition, ages ranging from 18 to 70 found that nutrition websites and google searches were identified as common sources. Kuske et al. [38] who undertook a systematic review of diabetes information-seeking behaviour found the internet is also often featured as a main source of information in people living with T2DM. While Meyfroidt et al [79] exploring the nutrition information sources preferred and used by older adults living with T2DM aged 60 and over found the internet was used as a source of nutrition information, Kalantzi et al [80] in their questionnaire study of attendees at a Greek hospital found that the internet is also more often made use of by younger adults living with T2DM than older adults. Similarly, Heuberger and Ivanitskay [54] found younger adults had a higher preference level for the internet.

Family and friends were found to be useful sources of nutrition information in seven studies [51,54,57,59,61,62,64]. McClinchy [67] also found family and friends valued sources of nutrition information in helping to manage T2DM and Vrinten et al. [69] identified that nutrition information behaviour was 'driven' by a desire for social interaction with others.

Gustafsson and Sidenvall [62] found that older adults carry embodied nutritional knowledge which impacted their food practices and that if they were preparing home cooked meals that they believed that this automatically meant they were eating a healthy diet. Embodied nutritional knowledge has been found to influence interaction with nutrition information and the consumption of a healthy diet amongst adults living with T2DM [67] and children [81]. Other studies have also found that the perception of home cooking (cooking from scratch) is a necessary part of a healthy diet [67,74,82].

A number of studies in this scoping review identified that access to information could influence food choices [53,55,59,60]. Whilst it is known that the provision of nutrition information does not automatically affect behaviour change [83], behaviour change models cite the provision of information as a key aspect in increasing knowledge and attitudes towards changing behaviour [84].

While one study did not find any gender differences [54], five studies found gender differences in use and preferences for different sources of nutrition information. Women were found to prefer practical information [58], television [57], radio, [57], friends [57,59], books [55], magazines [55] and food labels, [53]. Men were found to prefer the internet as a source of information [55]. However, this study found that a higher percentage of men preferred family and friends as sources of information than women [55]. The study undertaken by Jeruszka-Bielak et al. [55] was the only study involving an intervention (involving tailored advice on the Mediterranean diet and monthly dietetic counseling) which may have impacted the awareness of nutrition information sources in comparison to those studies not involving an intervention. Sbaffi and Rowley [85] found conflicting conclusions regarding gender preferences for online sources of information. The relationship between gender and nutrition information may be impacted by social norms. For example, Ek [86] and Hansen et al. [87] found that women were more likely to engage with nutrition information as they were more likely to have greater involvement in food provisioning. However, researchers are emphasizing the importance of taking a gender aware approach by collecting data on gender identity in order to ensure rigor and relevance [88].

Educational level impacted preferred nutrition information sources and the ability to make use of them. Higher education levels enabled the use of material requiring a higher reading age [57,58] and food labels as sources of information [53,60]. Those with higher levels of education were also more likely to possess critical appraisal skills enabling them to make use of a wider range of information [59]. Research has also found that possessing skills, literacy, and health literacy will impact whether adults can access certain sources of information [23,85] including digital technology [89]. Schroder et al. [89] identified 119 separate factors influencing interaction by older adults with digital technologies. They grouped these together into 9 themes: demographics and health status, emotional needs, knowledge and perception, motivation, social influencers, and technology functional features. Sbaffi and Rowley [85] found older adults have a preference for interpersonal consultations, tending not to trust online consultations, finding that access to online information is influenced by the format such as text size as well as level of education and prior experience with the internet [85]. Similarly, Turner et al. [65] exploring technology use amongst older adults found they are regular users of online technologies and that the barriers to use were related to the format (font size) and the preference for a website as opposed to an app.

Trust in a specific source of information was found to influence whether older adults would consult the source and whether the guidance impacted food choice. Other studies have found high reliance on information provided by health care professionals exploring health information practices in older adults and in adults with long term conditions particularly when the information was through interpersonal consultations [85]. For example, Hurst [37] undertook an exploratory qualitative study with older adults and found that the impact of age and the development of health-related conditions meant that health care professionals were the most common source of health information. The findings relating to credibility and trust in information provided by dietitians and doctors are supported by other academic work that illustrates the high levels of confidence older populations have in them [90], which may be enhanced when they have a medical diagnosis [91]. However, research suggests that this trust and therefore following advice from health care professionals may be outweighed by older adults' preferences and beliefs about what foods are good for them [76]. Also Gustafsson and Sidenvall [62] note that, while mistrusting sources, older adults did not feel as though they possessed the necessary skills or knowledge to critically appraise the information they came across.

Rueter et al. [64] identified that older adults felt that information found in the mass media was conflicting and confusing, an aspect which could be attenuated by the inclusion of evidence sources and dates of production [65]. Other works have also pointed to older adults' perceptions of health claims as being used as marketing tools, rather than being credible and reliable sources of nutrition information [92]. Elsewhere, it has been highlighted that such feelings of distrust have health implications and can affect consumers' abilities to maintain a healthy and balanced diet [93]. Understanding the health effects of mistrust amongst older consumers might provide valuable insights into further understanding the impact of nutrition information or lack of it on the causes of malnutrition.

### Strengths and limitations

The research team comprised subject and research experts [JM,AD,EB,TI], and members of the university's Public Involvement in Research Group with progress monitored by the public advisory group meetings held to support the study. The study involving searches across three databases was strengthened with an update in 2025. The team is confident that a wide range of search terms were incorporated; however, relevant search terms may have been omitted. While the research team acknowledges the relatively low number of included studies related to yield, this potentially indicates the small amount of research undertaken on this topic with free living older adults who do not have diagnosed medical conditions. The level of funding meant that the review process was limited to searching databases holding published research. As searching unpublished sources of studies was not part of the search process, this may have unintentionally excluded relevant research. While two of the databases (CINAHL and Scopus) include grey literature, these only extend to dissertations and conference proceedings. However, while dissertations would have been eligible for inclusion if they met the study design criteria, none were included, and conference proceedings were not eligible for inclusion. While the study is limited in generalizability to the UK and generally with one study involving participants from the UK [55], and just two studies recruited participants nationally [53,55], participants from 13 countries are represented in the current study. Overall, more women than men are represented as participants across the study. This may reflect social norms [86,87] and or be because women may be more interested in food-related issues so are more likely to take part in food related studies and they may have more nutrition knowledge than men and so may feel more confident in taking part in research about this topic [66].

### Further research and implications

The scoping review has identified a need for further research in this area, particularly in the UK. Future work could examine the role of, and advice given to older adults, about food and nutrition by health care professionals, as they were often a preferred and trusted source of nutritional information. As a wide range of sources are accessed and used, these are likely to reveal conflicting information and lead to mistrust in the information provided. Research is needed to explore the effectiveness of different sources of nutrition information in changing dietary behaviour. In order to identify the impact of sources on older adults, research should include a greater focus on information received on food choice.

The study findings will be of value particularly to nutrition and dietetic professionals working with older adults. Opportunities to assist in interpreting nutrition information sources such as food labels and exploring views and opinions about nutrition information accessed by older adults should be taken during consultations or nutrition communication events. An increased awareness of the possession of embodied nutrition knowledge, the value placed on practical sources of information, and those from trusted sources should inform nutritional consultations with older adults. When designing nutrition information resources, health care professionals should take the opportunity to make use of the inherent embodied knowledge and skills possessed by older adults in their development. The increased emphasis on communication within healthcare through digital and online formats mean there is a need for more research to co-develop and evaluate online age-friendly nutrition communication tools.

## Conclusions

Overall, this scoping review has identified the wide range of sources used by older adults, that the usability of these sources is impacted by levels of education, gender and by trust, and the potential for health claims on food labels to communicate nutrition information. There are opportunities for further research to explore the impact of nutrition information on food choice at a local level involving councils and charities as well as nationally by the government and National Health Service and assessing the impact of information sources on dietary outcomes. Older adults should be at the forefront of projects designing nutrition information for others in this cohort as they will be best placed to ensure that age-friendly nutrition communication tools accommodate sensory, cognitive, and literacy needs especially when information is provided online.

## Supporting information

**S1 File. Supporting information Food4years conference.**
(PPTX)

**S2 File. Supporting information Critical appraisal.**
(XLSX)

**S3 File. Supporting information PRISMA ScR checklist.**
(DOCX)

**S4 File. Supporting information Data.**
(XLSX)

## Acknowledgments

The research team is grateful for the support of the public advisory group in the monitoring of the study and Dr Rosalind Fallaize is acknowledged for their support in the design of the search process.

## Author contributions

**Conceptualization:** Jane McClinchy, Angela Dickinson, John Jackson, Amander Wellings.

**Data curation:** Jane McClinchy, Emily Barnes, Tai Ibitoye.

**Formal analysis:** Jane McClinchy, Angela Dickinson, Emily Barnes, Tai Ibitoye, John Jackson, Amander Wellings.

**Funding acquisition:** Jane McClinchy, Angela Dickinson.

**Investigation:** Jane McClinchy, Angela Dickinson, Emily Barnes, Tai Ibitoye, John Jackson, Amander Wellings.

**Methodology:** Jane McClinchy, Angela Dickinson, Emily Barnes, Tai Ibitoye, John Jackson, Amander Wellings.

**Project administration:** Jane McClinchy, Angela Dickinson.

**Resources:** Jane McClinchy, Angela Dickinson.

**Software:** Emily Barnes, Tai Ibitoye.

**Supervision:** Jane McClinchy, Angela Dickinson.

**Visualization:** Jane McClinchy.

**Writing – original draft:** Jane McClinchy, Angela Dickinson, Emily Barnes, Tai Ibitoye, John Jackson, Amander Wellings.

**Writing – review & editing:** Jane McClinchy, Angela Dickinson, Emily Barnes, Tai Ibitoye, John Jackson, Amander Wellings.

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
