## [Decision Letter · Decision Letter 0]

13 Oct 2025

Dear Dr. McClinchy,

Thank you for submitting your manuscript to PLOS ONE. After careful consideration, we feel that it has merit but does not fully meet PLOS ONE’s publication criteria as it currently stands. Therefore, we invite you to submit a revised version of the manuscript that addresses the points raised during the review process.

We look forward to receiving your revised manuscript.

Kind regards,

Dr Anh Nguyen

Academic Editor

PLOS ONE

2. During your revisions, please note that a simple title correction is required: Amend the title to remove ' Title: ' . Please ensure this is updated in the manuscript file and the online submission information.

“This work was supported by the Biotechnology and Biological Sciences Research Council Food for added life years: Putting research into action (Food4Years) grant.  [Award Reference RCP1007006, Funder Reference BB/W018349/1]. All authors contributed to the study design and agreed on the final article. TI and EB extracted the data, and JJ reviewed the data and confirmed study inclusion for the first search. JM and AD undertook the updated search. JM undertook the analysis and constructed the final article with AD. The research team is grateful for the support of the public advisory group in the monitoring of the study and Dr Rosalind Fallaize is acknowledged for their support in the design of the search process.”

“Initials: JM, AD,

Grant number: Award Reference RCP1007006, Funder Reference BB/W018349/1

Full name of each funder: Biotechnology and Biological Sciences Research Council Food for added life years: Putting research into action (Food4Years) grant.

URL: https://gtr.ukri.org/projects?ref=BB%2FW018349%2F1

The  funders did not play any role in the study design, data collection and analysis, decision to publish, or preparation of the manuscript.”

4. In the online submission form you indicate that your data is not available for proprietary reasons and have provided a contact point for accessing this data. Please note that your current contact point is a co-author on this manuscript. According to our Data Policy, the contact point must not be an author on the manuscript and must be an institutional contact, ideally not an individual. Please revise your data statement to a non-author institutional point of contact, such as a data access or ethics committee, and send this to us via return email. Please also include contact information for the third party organization, and please include the full citation of where the data can be found.

5. Please update your submission to use the PLOS LaTeX template. The template and more information on our requirements for LaTeX submissions can be found at http://journals.plos.org/plosone/s/latex.

Additional Editor Comments:

Thank you for submitting this interesting review. To improve clarity and methodological transparency, please consider the following points:

• Please clarify and justify the definition of “older people” and the chosen age threshold, noting variations across international frameworks (e.g., UN, WHO, OECD). This should be addressed in the Introduction.

• In lines 53–56, please briefly define “anorexia of ageing” for reader clarity, rephrase to avoid implying a direct causal link, and consider including psychological factors (e.g., depression, social isolation) for a more balanced explanation.

• The use of different screening tools (Rayyan vs Excel) and reviewer teams across the two search phases raises potential concerns about consistency and inter-rater reliability; further clarification is recommended.

• The paragraph in lines 478–493 reads as a list and lacks synthesis. Please integrate findings from prior studies and your review results more cohesively to improve flow and clarity.

• In the Acknowledgements section, please remove information about funding or competing interests. These should be reported separately in accordance with the Submission Guidelines.

Thank you for your careful attention to these points.

Reviewers' comments:

Reviewer's Responses to Questions

**Comments to the Author**

1. Is the manuscript technically sound, and do the data support the conclusions?

Reviewer #1: Yes

Reviewer #2: Yes

2. Has the statistical analysis been performed appropriately and rigorously?

Reviewer #1: Yes

Reviewer #2: N/A

3. Have the authors made all data underlying the findings in their manuscript fully available?

Reviewer #1: Yes

Reviewer #2: Yes

4. Is the manuscript presented in an intelligible fashion and written in standard English?

Reviewer #1: Yes

Reviewer #2: Yes

Reviewer #1: Sources of nutrition information used and preferred by older people: A scoping review

Peer Review Report

Title

The title can be rephrased, as the most suitable title for an academic journal should be based on clarity, scholarly tone, and alignment with scoping review conventions. My suggestion is:

“Sources and Preferences for Nutrition Information Among Older Adults: A Scoping Review”

This version is precise, avoids redundancy, and clearly signals the population, topic, and methodology. It uses “older adults,” which is preferred in academic and public health literature over “older people,” and it balances specificity with broad relevance for indexing and citation.

Abstract

The abstract clearly identifies the public health relevance of nutrition information for older adults, especially in the context of malnutrition risk. It outlines the scoping review methodology, including databases searched and inclusion criteria. The findings are succinctly presented, with attention to both source types and influencing factors. However, my suggestions are as follows:

1. Redundancy: Lines 14–16 repeat the aim unnecessarily. Consider condensing this into one clear sentence.

2. Clarity of Findings: The phrasing “8 studies showed…” and “6 studies showed…” is vague. Were these mutually exclusive? Did some studies report multiple sources?

3. Terminology: “Embodied knowledge” is mentioned without definition—this may confuse readers unfamiliar with the term.

4. Impact and Implications: The conclusion could more clearly articulate how future research or interventions might build on these findings.

Also, I suggested revising the Abstract. See the Suggested Revision of the Abstract below and you can add other information that are not included.

Revised Abstract -

A nutritionally adequate diet is essential for older adults to support healthy ageing and reduce the risk of malnutrition. With over a million older people in the UK affected or at risk, understanding where they source nutrition information is critical for designing effective public health interventions. This scoping review mapped existing studies on the sources and preferences for nutrition information among older adults. A comprehensive search of PUBMED, Scopus, and CINAHL (March 2023; updated February 2025) yielded 8936 records, of which 15 studies reporting on 14 research projects met inclusion criteria. Common sources included magazines, television, dietitians, general practitioners, family and friends, and personal experience (“embodied knowledge”), with educational level, gender, and trust influencing uptake and use. Further research is needed to assess the impact of these information sources and identify strategies to support older adults in making informed food choices that promote healthy ageing.

Introduction

1. Clarify the conceptual framework guiding the review—what theoretical lens informs the exploration of information sources?

2. Strengthen the rationale by linking the topic to broader public health goals such as SDG 2 (Zero Hunger) and SDG 3 (Good Health and Well-being).

3. Specify whether the review focuses on older adults in community settings, care homes, or both.

4. Consider integrating recent UK policy or demographic data to reinforce the urgency of addressing malnutrition in ageing populations.

Methods

1. Condense repetitive phrasing in some of the sub-sections and restate the aim and could be merged for clarity.

2. Provide more detail on inclusion/exclusion criteria—were qualitative studies included, and how was “preference” operationalized?

3. Clarify whether the review followed a formal framework (e.g., Arksey & O’Malley or PRISMA-ScR). Draw the PRISMA flow diagram again to reflect the Arksey & O’Malley PRISMA diagram.

4. Indicate whether any quality appraisal of included studies was conducted, even if not required for scoping reviews.

Results Presentation

1. Reorganize findings to distinguish between frequency of use and preference for sources—these are conceptually distinct.

2. Quantify overlaps—did some studies report multiple sources, and how were these counted?

3. Define “embodied knowledge” for clarity, as it may be unfamiliar to readers outside qualitative health research.

4. Consider presenting findings in a table or figure to improve accessibility and comparative insight.

Discussion

1. Deepen the analysis of how trust, gender, and education shape engagement with nutrition information.

2. Reflect on digital literacy and access—are older adults using online sources, and what barriers exist?

3. Compare findings with international literature to situate the UK context within broader ageing and nutrition trends.

4. Explore how cultural norms or generational attitudes may influence source preference and credibility.

Strengths and Limitations

1. Highlight the breadth of databases searched and the update in 2025 as strengths.

2. Acknowledge the limited number of included studies relative to the initial search yield.

3. Note potential publication bias—were grey literature or non-English studies excluded?

4. Discuss limitations in generalizability, especially if most studies focused on specific subgroups or regions.

Implications of Findings

1. Suggest how findings could inform tailored nutrition education strategies for older adults.

2. Recommend integrating trusted sources (e.g., GPs, family) into public health messaging campaigns.

3. Highlight the need for co-designed interventions that reflect older adults’ lived experiences and preferences.

4. Encourage future research to explore the effectiveness of different information sources in changing dietary behaviour.

Conclusion/Recommendations

1. Reframe the conclusion to emphasize actionable insights for practitioners and policymakers.

2. Specify which stakeholders (e.g., NHS, local councils, charities) could implement the recommendations.

3. Suggest developing age-friendly nutrition communication tools that accommodate sensory, cognitive, and literacy needs.

4. Reinforce the importance of ongoing evaluation to assess the impact of information sources on dietary outcomes.

Proofreading and Editing

1. Eliminate redundancy in aim statements and streamline sentence flow.

2. Correct minor grammatical inconsistencies (e.g., “nutrition5 status” should be “nutrition status”).

3. Ensure consistent terminology—use “older adults” or “older people” uniformly.

4. Improve transitions between sentences to enhance narrative cohesion.

References/In-Citations

1. Ensure all cited studies are current and relevant to the UK context or comparable ageing populations.

2. Include foundational scoping review methodology references (e.g., Levac et al., PRISMA-ScR).

3. Cite key public health reports on malnutrition in older adults (e.g., Age UK, NHS Digital).

4. Verify that all in-text citations match the reference list and follow the journal’s formatting style.

Reviewer #2: This is a well conducted scoping review and mionor revisions are recommended.

Abstract

Line 15- a word is missing?

Introduction

Make the first paragraph more general so it applies globally.

Line 48- ‘often lower’

Line 48/49 Note some nutrient requirements do change, for example vitamin D and calcium – reference needed for this statement

Line 54 Age-related anorexia is decreased appetite

Line 58- malnutrition is not normally defined this way- see Cederholm 2017 and more recent papers on GLIM; low bmi and weight loss are used when there are not more complete indicators of malnutrition in epidemiologic surveys e.g. Wolters 2019

The importance of understanding poor nutrition is important; you could cite the Malnutrition Awareness Tool that has been recently developed and validated in the Netherlands as an example of moving towards understanding older adults perceptions of malnutrition- suggest incorporating in the paragraph between lines 74 and 83

Line 103- clarify the them in this line to ‘food labels’

Line 185- describe the research team and public advisory group earlier in the methods- who were these people? What was their respective roles?

Clarify if extraction was done in duplicate or checked by a second author

Table 1- clarify the population- literature was included beyond UK, fix in first left column

Results

Table 2- Maccharle- spell out short forms in most right column

Table 2- Turner-final column ‘devices’ vs. devises’

Line 232 ‘other’ North America as Canada is in NA

Line 255-58- this statement belongs in the Discussion

Lines 297-300 review these sentences for overuse of ‘for example’ and incorrect punctuation

Line 343 ‘cross sectional’

Line 351 missing a %

Line 397- spelling of labels

Line 495 awkward sentence and missing final bracket on reference

Line 509 awkward sentence; the role of the advisory group was not clear in the methods

**Do you want your identity to be public for this peer review?** For information about this choice, including consent withdrawal, please see our Privacy Policy

Reviewer #1: **Yes:** Monica Ewomazino Akokuwebe, PhD

Reviewer #2: No

---

## [Author Response · Author response to Decision Letter 1]

4 Dec 2025

Dear Reviewers,

Thank you for your thorough and in-depth comments on our paper. We are grateful for the opportunity to make improvements. Please see below our response and or where you can find the amendments suggested:

Editor Comments:

,

• Please clarify and justify the definition of “older people” and the chosen age threshold, noting variations across international frameworks (e.g., UN, WHO, OECD). This should be addressed in the Introduction.

, We have revised to older adults inline with most of the included studies. We have justified the age range in the introduction lines 144-149

• In lines 53–56, please briefly define “anorexia of ageing” for reader clarity, rephrase to avoid implying a direct causal link, and consider including psychological factors (e.g., depression, social isolation) for a more balanced explanation.

, We have revised and simplified. (lines 45-46)

• The use of different screening tools (Rayyan vs Excel) and reviewer teams across the two search phases raises potential concerns about consistency and inter-rater reliability; further clarification is recommended.

, Revised see page 8 lines 206-213

• The paragraph in lines 478–493 reads as a list and lacks synthesis. Please integrate findings from prior studies and your review results more cohesively to improve flow and clarity.

, Revised-lines 559-575

• In the Acknowledgements section, please remove information about funding or competing interests. These should be reported separately in accordance with the Submission Guidelines.

, Removed

Reviewer 1 ,

Title

The title can be rephrased, as the most suitable title for an academic journal should be based on clarity, scholarly tone, and alignment with scoping review conventions. My suggestion is:

“Sources and Preferences for Nutrition Information Among Older Adults: A Scoping Review”

This version is precise, avoids redundancy, and clearly signals the population, topic, and methodology. It uses “older adults,” which is preferred in academic and public health literature over “older people,” and it balances specificity with broad relevance for indexing and citation.

, We have amended according to your suggestion

Abstract

The abstract clearly identifies the public health relevance of nutrition information for older adults, especially in the context of malnutrition risk. It outlines the scoping review methodology, including databases searched and inclusion criteria. The findings are succinctly presented, with attention to both source types and influencing factors. However, my suggestions are as follows:

, We have amended according to your suggestion

1. Redundancy: Lines 14–16 repeat the aim unnecessarily. Consider condensing this into one clear sentence.

, Revised according to your suggestion lines

2. Clarity of Findings: The phrasing “8 studies showed…” and “6 studies showed…” is vague. Were these mutually exclusive? Did some studies report multiple sources?

, Revised according to your suggestion

3. Terminology: “Embodied knowledge” is mentioned without definition—this may confuse readers unfamiliar with the term.

, We have added in a definition in the abstract line 13

4. Impact and Implications: The conclusion could more clearly articulate how future research or interventions might build on these findings.

, We have revised

Also, I suggested revising the Abstract. See the Suggested Revision of the Abstract below and you can add other information that are not included.

Revised Abstract -

A nutritionally adequate diet is essential for older adults to support healthy ageing and reduce the risk of malnutrition. With over a million older people in the UK affected or at risk, understanding where they source nutrition information is critical for designing effective public health interventions. This scoping review mapped existing studies on the sources and preferences for nutrition information among older adults. A comprehensive search of PUBMED, Scopus, and CINAHL (March 2023; updated February 2025) yielded 8936 records, of which 15 studies reporting on 14 research projects met inclusion criteria. Common sources included magazines, television, dietitians, general practitioners, family and friends, and personal experience (“embodied knowledge”), with educational level, gender, and trust influencing uptake and use. Further research is needed to assess the impact of these information sources and identify strategies to support older adults in making informed food choices that promote healthy ageing.

, Thank you, we have made incorporated your suggestions into a revision of the abstract.

Introduction

1. Clarify the conceptual framework guiding the review—what theoretical lens informs the exploration of information sources?

, We have explicitly referred to health information practice line 84 under the heading ‘Nutrition education resources used by older people’ and the use of the conceptual framework of health information practice in the aim. Line 148

2. Strengthen the rationale by linking the topic to broader public health goals such as SDG 2 (Zero Hunger) and SDG 3 (Good Health and Well-being).

, We have added a sentence referring to malnutrition being a global concern. Line 59

3. Specify whether the review focuses on older adults in community settings, care homes, or both.

, We have included the term free living in the aim, line 152, referred to the exclusion of studies that focus on those living in long term care facilities alone and reviewed the settings referred to in the studies to ensure that it is clear that the participants were free living although they may have been recruited whilst attending a hospital (see table 3 page 15).

4. Consider integrating recent UK policy or demographic data to reinforce the urgency of addressing malnutrition in ageing populations.

, We have added references to UK policy and demographic data 16,19-21

Methods Condense repetitive phrasing in some of the sub-sections and restate the aim and could be merged for clarity , We have removed repetitive phrasing, restated the aim line 159-161 and combined sections

Provide more detail on inclusion criteria-were qualitative studies included, and how was ‘preference’ operationalized? , We have referred to the need for eligible papers to refer to primary research in the text under the heading ‘criteria for inclusion’. Qualitative studies were included as indicated in table 2. Food preferences was one of the potential outcomes for eligible studies. See table 2

Clarify whether the review followed a formal framework (e.g. Arksey & O’Malley or PRISMA-ScR). Draw the PRISMA diagram again to reflect the Arksey & O’Malley PRISMA diagram. , We have revised the text to explain our use of a combination of Arksey and O’Malley and Levac methodologies. Lines 157-169. We have revised the PRISMA template to reflect that we used databases as our sources and and included table 1 to lay out the search process line 152

Indicate whether any quality appraisal of included studies was conducted, even if not required for scoping reviews , We did undertake a quality assessment. We have added a sentence indicating this. Lines 1171-173. The quality assessment is supplied in supporting information.

Results Presentation Reorganize findings to distinguish between frequency of use and preferences for sources-these are conceptually distinct , We have emphasised the difference in the concepts by revising the title to this section and revising the wording. Line 262-266

Quantify overlaps-did some studies report multiple sources and how were these counted? , This has been emphasised in the paragraph Sources relied on and preferred. Only 4 studies explored one source of information. Lines 263-266

Define ‘embodied knowledge’ for clarity, as it may be unfamiliar to readers outside qualitative health research , We have strengthened the explanation of embodied knowledge in the introduction. Lines 83-85

Consider presenting findings in a table or figure to improve accessibility and comparative insight , We have included an additional Table 6: Subthemes by study. Line 270

Discussion Deepen the analysis of how trust, gender, and education shape engagement with nutrition information. , We have enhanced these aspects trust lines 633-649, gender lines 597-613, education 615-631

Reflect on digital literacy and access-are older adults using online sources, and what barriers exist? , This has been revised as requested by the editor lines 559-575

Compare findings with international literature to situate the UK context within broader ageing and nutrition trends. , We have cited and referred to international literature references 68-91

Explore how cultural norms or generational attitudes may influence source preference and credibility , See lines 608-613

Strength and Limitations Highlight the breadth of databases searched and the update in 2025 as strengths , Line 666

Acknowledge the limited number of included studies relative to the initial search yield , Line 669

Note potential publication bias-were grey literature or non-English studies excluded? , Lines 674-678

Discuss limitations in generalizability, especially if most studies focused on specific subgroups or regions , Lines 678-681

Implications of Findings Suggest how findings could inform tailored nutrition education strategies for older adults. , Lines 702-709

Recommend integrating trusted sources (e.g., GPs, family) into public health messaging campaigns , Lines 704-705

Highlight the need for co-designed interventions that reflect older adults’ lived experiences and preferences , Lines 706-707

Encourage future research to explore the effectiveness of difference information sources in changing dietary behaviour. , Line 694

Conclusion/Recommendations Reframe the conclusion to emphasize actionable insights for practitioners and policymakers , We have reframed this focussing on involving older people and in the development and evaluation of online tools-line 719

Specify which stakeholders (e.g., NHS, local councils, charities) could implement the recommendations , We have specified national, local councils and charities

Suggest developing age-friendly nutrition communication tools that accommodate sensory, cognitive, and literacy needs. , We have included this line 721

Reinforce the importance of ongoing evaluation to assess the impact of information sources on dietary outcomes. , We have included this line 694

Proofreading and editing ,

Eliminate redundancy in aim statements and streamline sentence flow , We have reduced this section and simplified the aim lines 152-154

Correct minor grammatical inconsistencies (e.g., ‘nutrition5 status’ should be ‘nutrition status’) , We have corrected this error

Ensure consistent terminology-use ‘older adults’ or ‘older people’ uniformly. , We have revised to older adults apart from where a specific study uses the word in their aim or title

Improve transitions between sentences to enhance narrative cohesion , We have rewritten to ensure transitions

References/In-Citations ,

Ensure all cited studies are current and relevant to the UK context or comparable ageing populations , These have been checked

Include foundational scoping review methodology references (e.g. Levac et al., PRISMA-SCR). , References 42 and 43

Cite key public health reports on malnutrition in older adults ( e.g. Age UK, NHS Digital). , References 16 and 19

Verify that all in-text citations match the reference list and follow the journal’s formatting style , We have checked the references

Reviewer 2. This is a well conducted scoping review and minor revisions are recommended ,

Abstract Line 15-a word is missing? , We have revised the abstract

Introduction Make the first paragraph more general so it applies globally , Line 29

Line 48- ‘often lower’ , Revised

Line 48/49 Note some nutrition requirements do change for example vitamin D and calcium-reference needed for this statement , We have used SACN which is the latest guidance on nutrition requirements for the UK reference 5 p29

Line 54 Age-related anorexia is decreased appetite , We have revised this line 39

Line 58 –malnutrition is not normally defined this was-see Cederholm 2017 and more recent papers on GLIM; low bmi and weight loss are used when there are not more complete indicators of malnutrition in epidemiological surveys e.g. Wolters 2019 , We have revised this and replaced the Cederholm reference with the reference referring to the two-stage approach. Lines 43-46

The importance of understanding poor nutrition is important; you could cite the Malnutrition Awareness Tool that has been recently developed in the Netherlands as an example of moving towards understanding older adults perceptions of malnutrition-suggest incorporating in the paragraph between lines 74 and 83 , We have incorporated this- line 67-69 ref 22

Line 103-clarify the them in this line to ‘food labels’ , Food labels, we have revised this

Line 185-describe the research team and public advisory group earlier in the methods-who were these people? What was their respective roles? , We have described this in the introduction lines 137-149

Clarify if extraction was done in duplicate or checked by a second author , Extraction was undertaken by each assessor confirmed with each other and checked by the lead author line -224

Table 1-clarify the population-literature was included beyond the UK, fix in the first left column , Revised: This was expanded to international as few studies involved older people who lived in the UK.

Results ,

Table 2-Maccharle-spell out short forms in the most right column , Revised

Table 2-Turner-final column ‘devices’ vs. ‘devises' , Revised

Line 232 ‘other’ North America as Canada is in NA , Revised North America to United States of America line 232

Line 255-58-This statement belongs in the discussion , We have moved this to strengths and weaknesses

Lines 297-300 review these sentences for overuse of ‘for example’ and incorrect punctuation , We have revised this -line 373

Line 343 ‘cross sectional’ , We have corrected this -line 411

Line 351 missing a % , We have corrected this -lines 291-293

Line 397-spelling of labels , We have corrected this line 446

Line 495 awkward sentence; the role of advisory group was not clear in the methods , We have clarified -lines 137-149

Use PACE for figure upload-see revision email for how to do this , We have undertaken this

,

,

---

## [Editor Report · Decision Letter 1]

30 Dec 2025

Sources and preferences for nutrition information among older adults: A scoping review.

PONE-D-25-20562R1

Dear Dr. McClinchy,

We’re pleased to inform you that your manuscript has been judged scientifically suitable for publication and will be formally accepted for publication once it meets all outstanding technical requirements.

Kind regards,

Dr Anh Nguyen

Academic Editor

PLOS One

---

## [Editor Report · Acceptance letter]

PONE-D-25-20562R1

PLOS One

Dear Dr. McClinchy,

I'm pleased to inform you that your manuscript has been deemed suitable for publication in PLOS One. Congratulations! Your manuscript is now being handed over to our production team.

Kind regards,

on behalf of

Dr. Anh Nguyen

Academic Editor

PLOS One